# Shared characteristics of intervention techniques for oral vocabulary and speech comprehensibility in preschool children with co-occurring features of developmental language disorder and a phonological speech sound disorder: protocol for a systematic review with narrative synthesis

Lucy Rodgers [ID],[1,2] Nicola Botting,[2] Martin Cartwright,[3] Sam Harding [ID],[4] Rosalind Herman[2]

For numbered affiliations see end of article.

**Correspondence to**
Lucy Rodgers;
lucy.rodgers@city.ac.uk

## ABSTRACT

**Introduction** Evidence suggests that over one-third of young children with developmental language disorder (DLD) or speech sound disorder (SSD) have co-occurring features of both. A co-occurring DLD and SSD profile is associated with negative long-term outcomes relating to communication, literacy and emotional well-being. However, the best treatment approach for young children with this profile is not understood. The aim of the proposed review is to identify intervention techniques for both DLD and SSD, along with their shared characteristics. The findings will then be analysed in the context of relevant theory. This will inform the content for a new or adapted intervention for these children.

**Methods and analysis** This search will build on a previous systematic review by Roulstone *et al* (2015) but with a specific focus on oral vocabulary (DLD outcome) and speech comprehensibility (SSD outcome). These outcomes were identified by parents and speech and language therapists within the prestudy stakeholder engagement work. The following databases will be searched for articles from January 2012 onwards: Ovid Emcare, MEDLINE Complete, CINAHL, APA PsycINFO, Communication Source and ERIC. Two reviewers will independently perform the title/abstract screening and the full-text screening with the exclusion criteria document being revised in an iterative process. Articles written in languages other than English will be excluded. Data will be extracted regarding key participant and intervention criteria, including technique dosage and delivery details. This information will then be pooled into a structured narrative synthesis.

**Ethics and dissemination** Ethical approval is not needed for a systematic review protocol. Dissemination of findings will be through peer-reviewed publications, social media, and project steering group networks.

**PROSPERO registration number** CRD4202237393.

---

### STRENGTHS AND LIMITATIONS OF THIS STUDY

⇒ This protocol follows the Preferred Reporting Items for Systematic Review and Meta-Analysis Protocols guidelines.
⇒ Electronic databases spanning medicine, education and psychology will be searched.
⇒ Electronic databases in languages other than English will not be searched.
⇒ Meta-bias(es) within the literature cannot be fully controlled.
⇒ The level of detail within intervention reporting, as per the TIDieR (Template for Intervention Description and Replication) guidelines, has the potential to vary among studies.

---

## INTRODUCTION

Within the field of child language disorders, there are often overlapping or co-occurring difficulties which create unique patient experiences. Yet, while there is ample literature on treatment for singly occurring difficulties, there is a notable gap in evidence for treating children with co-occurring disorders. This review focuses on intervention for children who have co-occurring features of both developmental language disorder (DLD) and speech sound disorder (SSD).

### Co-occurring DLD and SSD

An estimated 7.58% of 4-year olds present with features of a DLD.[1] DLD is characterised by idiopathic difficulties in using and understanding spoken language.[2] One

feature is limited vocabulary development,[2] which has a known association with childhood temper tantrums/mental health, and later language and literacy skills.[3 4] Such features of DLD may co-occur with a speech sound disorder (SSD); that is, difficulties in producing speech sounds.[5] An estimated 3.4% of 4-year olds have SSD.[6] One of the most devastating impacts of SSD is the impact on a child's ability to make themselves understood to others in everyday life.[7] The term for this is speech comprehensibility.[8] A related term, speech intelligibility, refers to the acoustic–phonetic decoding of utterances, and is very closely related to speech comprehensibility as both are linked to the functional use of speech. As with limited vocabulary, poor speech comprehensibility/intelligibility within the early years have also been associated with negative longer-term outcomes, including persisting speech difficulties[9] and poor literacy skills.[10 11] Although it is typical for very young children not to be fully understood to those around them as their speech develops, by 4 years of age a child would typically be at least 50% intelligible.[12]

Thirty six per cent of 4-year olds with idiopathic SSD also have oral (ie, expressive-spoken) language features of DLD.[6] This high rate of co-occurrence is in keeping with historical research in the area,[13] as well as study data from clinical caseloads.[14] The combined impact of co-occurring features of DLD/SSD is twofold; for example, for a child with limited oral vocabulary and speech comprehensibility, not only are they unable to use many words, but the limited words they do have will not be understood to others within their daily lives. It therefore may be unsurprising that co-occurring phonological DLD/SSD features in early childhood are associated with negative long-term outcomes relating to literacy[15 16] and communication,[17 18] with downstream consequences for quality of life[18 19] and emotional well-being.[20] Consequently, access to effective and appropriately targeted intervention for children with this profile is crucial.

Phonological SSDs are the most frequently presenting SSD subtype,[5] and occur when a child has difficulties with manipulating the different sound contrasts (phonemes) which are needed to form words.[21] There are different types of phonological SSDs, including consistent phonological disorder (where the child makes consistent sound omissions or substitutions) and inconsistent phonological disorder (where these errors have no consistent pattern).[21] Research highlights a known link between DLD and phonological SSDs, as both disorders are underpinned by shared linguistic deficits.[2] This overlap is represented in the seminal CATALISE DLD consensus paper.[2] In contrast to phonological SSDs, the CATALISE authors' speech, language and communication needs diagram highlights how other SSD subtypes, such as motor-based SSDs like dysarthria, have a less marked overlap with DLD. Although non-phonological SSDs such as articulation disorder and childhood dyspraxia of speech (CAS) could also be idiopathic, other non-phonological SSDs often are not. Due to their significant overlap with DLD

which has no known causation, this review will focus on phonological SSDs which are also idiopathic in nature.

Speech, language and communication needs are illustrated in figure 1.[2]

The overlap between language and phonological SSDs is further supported by studies on the speech and language development of young children, where complex and bidirectional relationships between the development of individual sounds (phonology) and words (the lexicon) have been identified.[22 23] For example, the first words of young children primarily consist of the speech sounds already established within their emerging phonological inventory.[23] This relationship between phonology and the lexicon may have implications for intervention with children with co-occurring features of DLD and a phonological SSD. For example, growth in vocabulary and/or the strengthening of phonological representations has the potential to impact speech and vocabulary development concurrently through a process known as 'lexical restructuring'.[24] A further psycholinguistic theory of potential relevance is the speech processing model,[25] which suggests that individual children with co-occurring features of DLD/a phonological SSD may have difficulties at one or more levels of speech processing, rather than just with phonological representations alone. Such theories are important within interventions for co-occurring DLD/phonological SSD as they can be used to inform intervention content and delivery.

## Current interventions for preschool co-occurring DLD/SSD

Although this overlap exists between DLD and phonological SSDs, there is currently a paucity of theoretically informed interventions which have been specifically developed for this group.[26] Additionally, intervention studies within existence primarily target morphosyntactic aspects of expressive language, alongside accuracy of speech sound production.[26 27] However, for younger children with this profile, and children whose features of DLD are more severe, building vocabulary is typically targeted in speech and language therapy prior to morphosyntax.[28]

'Child Talk'[29] was a large National Institute of Health Research (NIHR) funded mixed methods programme of work, including a systematic review. This involved investigating the use of early years' speech and language therapy interventions. The findings led to the specification of (1) a typology of early years' speech and language therapy (SLT) intervention, (2) key intervention ingredients for each typology theme. A technique can be described as 'the specific teaching behaviours/actions thought to effect change'.[30] The findings highlighted that for children with co-occurring features of DLD/SSD, clinicians often adapt existing interventions by selecting and combining different techniques. This enables them to use their knowledge and experience to provide the best treatment that they can.[29 31]

Although our knowledge of what techniques work best for children with this profile is limited, techniques identified might be related to underlying theories of potential

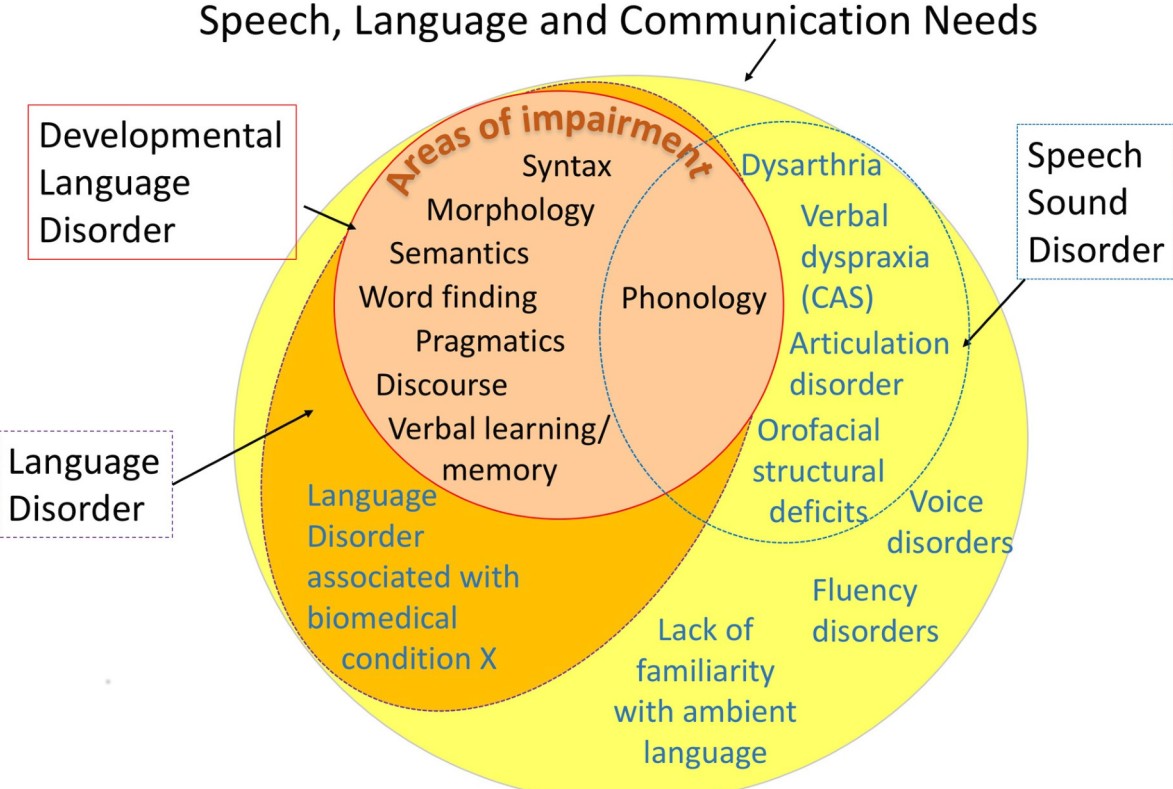

**Figure 1** Speech, language and communication needs.[2]

relevance. For example, language modelling is typically linked to growth in expressive language.[32] However, based on the lexical restructuring hypothesis, it is hypothesised that the subsequent impact of this language growth on the accuracy and segmentation of the child's phonological representations could also influence their phonological speech sound production.[24] Using various techniques to 'build things into play' was also highlighted in Child Talk. Theoretically, this is supported by the latest research on the brain basis of speech and language learning, which indicates that learning best takes place within interactions which are meaningful for the child.[33 34] Romeo *et al* (2018) found that Broca's area of the brain became activated in response to a child being exposed to meaningful back and forth interactions, rather than in response to passively 'hearing' words. Considering this, it is hypothesised that this technique supports speech and language learning through capitalising on the child's heightened attention and motivation during the play activity.

These considerations highlight a valuable opportunity for an intervention specific to this clinical group to be developed, using techniques which can be supported by relevant theory. Due to the current paucity of evidence, the associated negative impact of this co-occurring profile on long-term outcomes, and the high level of presentation on clinical caseloads, there is an urgent need for such intervention development to take place. The first stage in this development would be to conduct a systematic review to identify potential techniques of relevance.

### Broader context: an intervention development study

The proposed review updates the systematic review findings from 'Child Talk'[29] while refining the focus to techniques within interventions for children with features of DLD or a phonological SSD. Techniques will be extracted from included studies and then analysed in relation to shared characteristics and underpinning theory. The synthesis will then be used to inform the content of a new intervention which is being developed for young children with co-occurring features of DLD/phonological SSD.

Both DLD and SSD are heterogenous disorders,[2 21] and therefore have a range of associated outcomes. This review, and body of intervention development work it is a part of, will focus exclusively on the outcomes of oral vocabulary (DLD outcome) and speech comprehensibility (SSD outcome). This is due to the aforementioned impact of such difficulties on the everyday lives of young children; this decision is also elaborated on in the 'patient and public involvement' section of this paper.

Based on the dose form framework,[30] shared characteristics for DLD/phonological SSD intervention techniques may include similarities in:

1. *Who* delivers the technique; for example, is it the parent, clinician or both?
2. *Where* the technique is delivered; for example, at home, nursery, clinic or a combination of these?
3. The nature of *technique delivery*; for example, is the activity presented in an adult led structured game, play, everyday routines or a combination of these?

Underpinning theory may relate to:
1. The lexical restructuring hypothesis.[24]
2. Psycholinguistic models of speech and language development; such as the speech processing model.[25]
3. The neural basis for speech and language development; for example, the role of meaningful interactions within language learning[33]

## Objectives

The overarching aim of the review is to bring together intervention techniques from DLD and phonological SSD interventions. The objectives within this are to:
1. Identify the shared core characteristics of the techniques; this includes the deliverer, place of delivery, format of delivery and nature of delivery (*eg, child or adult led*)
2. Compare and synthesise the shared core characteristics of the techniques in relation to underlying theory
3. Establish the best available evidence for interventions that incorporate these core characteristics of the intervention techniques.

## Research questions

1. What are the shared core characteristics of intervention techniques in preschool interventions targeting speech comprehensibility and/or oral vocabulary?
2. How do these shared core characteristics relate to underlying theory?
3. What evidence is there for the effect of interventions that incorporate these core characteristics of intervention techniques?

## METHODS AND ANALYSIS
### Study registration

In accordance with the guidelines, our systematic review protocol has been registered with the International Register of Systematic Reviews (PROSPERO) on 16 December 2022. In the event of any amendments to methodology set out below, the date of each amendment will be accompanied by a description of the change and the rationale in either the PROSPERO register and/or the final results paper.

### Eligibility criteria

The eligibility criteria stated below are in line with the criteria from the original 'Child Talk' systematic review[29] with amendments according to the objectives of the current review. Most importantly, this review will focus specifically on the 'expressive language' and 'speech' themes generated from their initial typology of early years' SLT interventions, as these themes encompass the two outcomes for which we are seeking to identify techniques.

### Study designs

Included studies must report on an empirical evaluation of the effectiveness of an intervention. To ensure we identify all relevant literature, a range of study designs will

be included. These include randomised control trials, experimental and quasi-experimental studies, within subjects designs (eg, pre–post studies) and case studies (which may include multiple baseline or other systematic manipulation of the intervention). Studies which report on single time point (eg, cross-sectional studies) will be excluded. Studies focusing on efficacy, including laboratory-based training, will not be excluded if all other inclusion criteria are met. This is because information on the efficacy of speech/language learning techniques can be gleaned from these studies, although careful consideration will be given to how these results are integrated into the narrative analysis (further information on this is provided under 'data synthesis').

### Population

To capture the age group most typically seen within clinical services, 80% of children within included studies must have been aged between 2:0 and 5:11 years. Additionally, although this review is part of a wider intervention development study for children aged 3 and 4 years, an expanded age range within this review will help to ensure that techniques of potential relevance will be captured. The children within included studies must have presented with phonological speech production difficulties and/or difficulties relating to oral vocabulary, with all subtypes of phonological SSD included (eg, consistent and inconsistent phonological disorder, phonological delay). These difficulties may be identified by standardised assessments such as the Preschool Language Scale,[35] parental and/or professional observation reports such as the intelligibility in context scale[36] and/or probes. Probes may also be used to assess progress through the repeated measurement of the dependent variable before, during and after the intervention. As already observed in the literature, common probes within speech and language therapy interventions may include a selection of words containing the child's targeted speech sound/s or vocabulary.[37 38] In keeping with the aforementioned diagnostic description within CATALISE,[2] included papers must state that the participants' needs had no obvious cause, that is, excluding children with neurodevelopmental differences that have a known association with speech and/or language development, such as autism or cerebral palsy. Due to the challenges in diagnosing DLD in very young children,[2] and in order to maximise the identification of potentially relevant intervention techniques, studies will be included where the child does not have a formal diagnosis of DLD but is a late talker.

### Interventions

We will include studies reporting on interventions delivered in any setting (eg, home based, clinic) or format (eg, face-to-face, online). The deliverer may be a speech and language therapist, speech and language therapy assistant or equivalent professional (including education staff), and the intervention may involve professionals

training up others (eg, parents) to deliver some or all of the intervention.

## Comparators

Comparators for included studies may be a control without an intervention (including multiple baseline and within subjects designs) or an alternative experimental group (ie, intervention comparison).

## Outcomes

Included papers must measure the effectiveness of the intervention on (1) oral vocabulary, and/or (2) speech comprehensibility. These outcomes must be evaluated via standardised assessment, probes and/or observational ratings or scales.

If composite speech and language assessments are used, studies must report on the separate subtest results for oral vocabulary and/or speech comprehensibility to be included.

Studies with only syntactic measures of language change will be excluded; this includes mean length of utterance in morphemes. However, they will be included if a proximal measure of vocabulary change is used alongside syntactic measures, such as the number of different words. Other outcome measures related to oral vocabulary might include parent report instruments and type-token ratios from language samples.

Speech comprehensibility is the SSD outcome in focus. As previously mentioned, comprehensibility and intelligibility are overlapping but differing constructs, with a shared focus on functional human communication.[8] Therefore, we will also include studies with an outcome of improved speech intelligibility as a proxy for comprehensibility. This was deemed more suitable than using measures such as PCC (Percentage of Consonants Correct) as a proxy for comprehensibility, with their focus being on speech accuracy. Due to the very recent consensus in terminology, measures for comprehensibility might include measures with 'intelligibility' within their title, such as the 'Intelligibility in Context' Scale (ICS), which is becoming increasingly used in SSD intervention research.[36 38] In the ICS, parents are asked to rate their child's speech comprehensibility according to the communication partner they are with within their everyday environments, thus providing high ecological validity. We will also include non-parent/ significant rated measures when looking at comprehensibility. For example, orthography-based approaches where raters are not known to the child.[39] Given the recent clarification on consensus on intelligibility versus comprehensibility,[8] it could be argued that this approach falls between the two, with there being less focus on who the speaker's communication partner is, and the wider context which the interaction takes place in.[8] Regardless, such studies will still be included, as the results still relate to functional human communication.

## Information sources

The search will be conducted in Ovid Emcare, MEDLINE Complete, CINAHL, APA PsycINFO, ERIC and Communication Source. These sources have been selected as they encompass the fields of health (medical, nursing and allied health professions), speech and language therapy, education and psychology and have been successfully used in previous reviews in the field.[26 40] To support literature saturation, supplementary search methods will be employed; this includes screening the reference lists from prominent reviews in the field post 2012.[41 42] We have selected reviews from 2012 onwards due to the original search going up to this date.[29] Reference lists from included papers within the current search will also be screened for potential study eligibility. Forward citation searches in Web of Science (using the core collection) will also be carried out, with additional searches in Scopus if the titles are not available in Web of Science.

Due to resource constraints, articles written in languages other than English will be excluded. However, articles written in English where the participants speak languages other than English will be included. Additionally, grey literature searching will be confined to the inclusion of theses/dissertations, via the databases stated above. Thesis/dissertations have been selected as although the original review[29] included a range of grey literature, thesis/dissertations were the only grey literature sources which contributed studies within the final included papers. In keeping with the original review, thesis/dissertations will only be included when a corresponding journal article cannot be found for the study.

## Search strategy

Together with support from a specialist librarian, we will conduct an update of the original 'Child Talk' systematic review,[29] searching articles from January 2012 to the present day. One of the researchers (SH) undertaking the current search also led on the original review. Relevant studies from the original 'speech' and 'expressive language' typology themes within the original 'Child Talk' review have already been located by reviewing the recorded outcomes for each study as stated on the original data extraction spreadsheet. Out of 41 papers from the 'speech' theme, 2 were found to address the outcome of comprehensibility/intelligibility. From the 30 papers within the 'expressive language' theme, 12 were found to include oral vocabulary as an outcome. These 14 papers will be further screened at stage 2 of the screening process (full-text stage, outlined below).

The original review search strategy[29] has been updated for the current review, accounting for advances in terminology, for example, consensus on the term 'Developmental Language Disorder'.[2] The original 'Child Talk' search encompassed a broader range of speech and language outcomes, therefore the search terms for the current review have been adjusted to focus on our two specific outcomes of interest; oral vocabulary and speech comprehensibility. The updated search strategy was initially reviewed by two independent postdoctoral researchers in the field and adjusted as needed, for example, adding in the term 'specific language

impairment', which may be relevant to older papers in the search. For the revised search strategy draft for each database, please see online supplemental material 1.

### Study records, selection and data collection process

Search results will initially be imported into RefWorks, where duplicates will be removed by the first author (LR). The remaining articles will then be uploaded to the Covidence systematic review management database.

Initially, the first author (LR) will trial the exclusion guidance criteria document on 30 papers. For the initial draft of this exclusion criteria guidance document, please see online supplemental material 2. The 30 papers will be randomly selected using a random number generator. These 30 papers will then be reviewed by a second reviewer. The reviewers will then meet to discuss discrepancies and make amendments to the exclusion guidance document if needed.

The screening and data extraction will be carried out as follows:

### Title/abstract screening

The full set of titles/abstracts will be screened by the first author (LR). If uncertainty arises about how to apply eligibility criteria to a specific paper, these articles will be discussed with a member of the review team (who is not involved in the formal screening process). This may then lead to further revisions to the exclusion criteria document. Following this, a second independent reviewer (SH) will independently screen the titles/abstracts. Any disagreements, and how these may relate to the exclusion guidance document, will be discussed in consensus meetings. Any disputed articles will then be re-screened should alterations have been made to the exclusion criteria document. If disagreement is not caused by confusion over the exclusion criteria document, and consensus cannot be reached through discussion, a third reviewer will be consulted.

### Full paper screening

At the full-text screening stage, two reviewers (LR and SH) will independently appraise all of the remaining articles for inclusion, following the iterative process as outlined for stage 1.

To enable transparency of the reliability of screening at stages 1 and 2, Cohen's κ for these stages will be reported in the final paper.

### Risk of bias/internal validity

Retained studies will then undergo assessment of internal validity by two independent reviewers (LR and SH). The reviewers will have regular consensus meetings, after independently assessing up to four papers at a time, to resolve potential conflicts. If disagreements persist, a third reviewer will be involved. Disagreements that arise (including those that have been resolved) will be recorded and reported in the final paper.

For the Physiotherapy Evidence Database PsychBITE (PEDro-P),[43] papers with a rating of 6 and over will be retained for data extraction. This aligns with the original review.[29] On the Risk of Bias in N-of-1 Trials (RoBiNT) scale, included studies will be rated as *fair* or above.[44]

### Data extraction

The first author (LR) will extract data from the first 25% of studies. These will be randomly selected using random number generation. A second extractor (SH) will then independently extract data from the same studies. The two extractors will then meet to discuss potential discrepancies, and to update the data extraction form if needed. Following this, the first author (LR) will extract the remainder of the data.

### Data items

Data will be sought regarding general study information (eg, date; study type; location; participant numbers), population characteristics (eg, male/female; age; speech/language profile-including phonological SSD subtype), intervention techniques (eg, dosage; underpinning theory and justification given by the authors) and reported impact on the outcome of interest. Data on reported participant socioeconomic status background will also be obtained, due to this being a known risk factor within developmental speech and language disorders.[45] We will also collate information on the number of languages spoken by the participants, as well as reported ethnicities, with this being a potential factor for the external validity of findings (ie, relevance to 'everyday' clinical practice).

The Template for Intervention Description and Replication (TIDieR) will be used as a framework to guide the extraction process,[46] combined with the speech and language therapy specific 'Dose form Framework and Definitions', based on the work of Warren and colleagues[47]; this has been applied in other reviews specific to paediatric speech and language therapy intervention.[41] Details of techniques will be extracted regarding intervention contexts (eg, the overarching activity the technique is presented in), method of instruction (eg, who delivers the technique, where and when) and technique dosage (dose frequency and dose duration). All reported dosage information will be extracted in order to allow for variation in study design; most notably, studies which target both oral vocabulary and speech comprehensibility concurrently.

### Outcomes and prioritisation

The two outcomes (oral vocabulary, speech comprehensibility) are of equal interest within this review, regardless of whether they are primary or secondary outcomes within the included studies.

### Risk of bias in individual studies

Individual studies will be assessed for internal validity. To encompass the range of study designs included within this review, we will use the PEDro-P.[43] Specifically, for single case experimental designs, the RoBiNT scale will be used.[44]

## Data synthesis

### Quantitative data

Overarching details for each included study, including the individual internal validity ratings, will be given in the first table. Two summary graphs will also be presented to convey the percentage of overall ratings from the PEDro-P and RoBiNT scales. The frequency of techniques within the included papers will be presented either numerically within a table, or within a graph or chart if this is deemed more suited to the data collected. We will be guided by the synthesis without meta-analysis in systematic review guidelines[48] and will report on the direction of effect of the interventions, using vote counting with a sign test if appropriate.

### Qualitative data

A description of the identified techniques will be presented in a table, including details regarding how they were operationalised, based on TIDieR[46] and the dose form framework.[30 47]

### The narrative synthesis will include sections on

1. Similarities and differences (including shared core characteristics) between techniques used for the different outcomes
2. Patterns of technique dosage and delivery across the interventions
3. How the similarities and differences (including shared core characteristics) in techniques relate to underlying theory. Depending on findings, this section will be broken down into subsections focusing on each theory of interest, potentially including (but not necessarily limited to):
   – The lexical restructuring hypothesis.
   – The speech processing model.
   – The neural basis for speech and language development.
4. The effectiveness of interventions which contain these techniques/shared core characteristics of techniques.

If relevant, any observed differences between interventions for different phonological SSD subtypes will be incorporated into the narrative synthesis, or given in an additional section if deemed to be more appropriate to the data found.

In the event of laboratory-based training studies meeting the final inclusion criteria, this data will be presented on a separate table. Additionally, within the narrative synthesis itself they will not be directly compared with the effectiveness studies. Instead, they will be used to support any potential theory building arising from the synthesis.

If challenges are identified regarding gaps and quality in the knowledge base, this will also be explored within the results and discussion section of the corresponding results paper.

## Patient and public involvement

According to the James Lind Alliance, knowing how to best select communication strategies according to a child's individual profile is the second most important recommendation for research.[49] This is strongly in keeping with the aims of this review, and highlights the broader relevance of this work.

For our wider intervention development study, outcomes were prioritised by clinicians and parents of preschool children with DLD/SSD within prestudy Patient and Public Involvement and Engagement (PPIE) work.[50 51] They identified the outcomes of increasing (1) oral vocabulary (*DLD outcome*), and (2) speech comprehensibility (*SSD outcome*). This provides further support focusing on techniques that directly target oral vocabulary and speech comprehensibility.

In keeping with the integral role of PPIE throughout, a newly formed project PPIE steering group will provide input at key points in the review process. This is a diverse group consisting of parents, speech and language therapists, a person with DLD, a specialist early years educator, a bilingual/multilingual educational family support worker and a clinical equality, diversity and inclusion expert. During the review, they will be involved with:

1. Reviewing the content for the data extraction form (online supplemental material 3), prior to the data extraction phase.
2. Identifying what data has the most relevance in the 'real world', with these potentially informing recommendations within the final paper.
3. Defining and agreeing key messages to take from the review, and dissemination through the steering group networks.

Steering group input will be recorded and reported in the final article, in accordance with the Guidance for Reporting Involvment of Patients and the Public (GRIPP) 2 reporting criteria short form.[52]

## Meta-bias(es)

It is important to acknowledge that meta-bias, including reporting and publication bias, is present within all aspects of health research. Although it is not possible to completely control for such bias, we will:

1. Establish if the protocol for each study was published before recruitment for participants commenced (where possible).
2. Compare the outcomes and results sections of the published report when a protocol is available (for when considering selective reporting bias).
3. Assess potential publication bias through the inclusion of prioritised grey literature (thesis/dissertations).

## Confidence in cumulative evidence

Confidence within the evidence as a whole will be based on the summary of the internal validity, as presented in the two summary tables (see data synthesis section). We will also acknowledge and discuss key factors relating to meta bias, and how the review findings should be interpreted based on this.

## Ethics and dissemination

As a systematic review this study does not warrant ethics board approval. Findings will be disseminated through peer-reviewed publications, social media, and project steering group networks.

**Author affiliations**
¹Children's Speech and Language Therapy, Sussex Community NHS Foundation Trust, Brighton, UK
²Department of Language and Communication Science, City University of London, London, UK
³Department of Health Services Research and Management, City University of London, London, UK
⁴Bristol Speech and Language Therapy Research Unit, North Bristol NHS Trust, Westbury on Trym, UK

**Acknowledgements** The authors would also like to acknowledge and thank Dr Pauline Frizelle (University College Cork) and Dr Helen Stringer (Newcastle University) for providing expert independent reviews of the initial terms for the search strategy.

**Contributors** LR led on this work and independently developed an initial draft of the manuscript (and appended documents). RH, NB, MC and SH suggested amendments after reviewing this and three subsequent re-drafts by LR. LR completed the fourth and final version of the manuscript, which was reviewed and agreed by all of the authors. LR led on manuscript revisions following peer review and all authors agreed to the final submitted version.

**Funding** This research was funded in whole, or in part, by the Wellcome Trust (Grant number 223500/Z/21/Z). For the purpose of open access, the author has applied a CC BY public copyright licence to any Author Accepted manuscript version arising from this submission.

**Competing interests** None declared.

**Patient and public involvement** Patients and/or the public were involved in the design, or conduct, or reporting, or dissemination plans of this research. Refer to the Methods section for further details.

**Patient consent for publication** Not applicable.

**Provenance and peer review** Not commissioned; externally peer reviewed.

**ORCID iDs**
Lucy Rodgers http://orcid.org/0000-0001-6585-7206
Sam Harding http://orcid.org/0000-0002-5870-2094

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
