## [Reviewer comments · BMJ Open]

ARTICLE DETAILS

TITLE (PROVISIONAL)	The shared characteristics of intervention techniques for oral vocabulary and speech comprehensibility in pre-school children with co-occurring features of developmental language disorder and a phonological speech sound disorder: protocol for a systematic review with narrative synthesis
AUTHORS	Rodgers, Lucy; Botting, Nicola; Cartwright, Martin; Harding, Sam; Herman, Rosalind

VERSION 1 – REVIEW

REVIEWER	Joanne Cleland University of Strathclyde, Psychological Sciences and Health
REVIEW RETURNED	20-Feb-2023

GENERAL COMMENTS	Thank you for asking me to review this protocol. I think this is a useful topic and the resulting systematic review will be of benefit to speech and language therapists. My main comment is that some of the terminology and rationale needs tidied up. I think the introduction would benefit from further information- much of which is covered in the method, but needs to be addressed earlier. In the intro I think it would be helpful to be clearer if the review will consider all types of SSD. You could refer to the Catalise Venn diagram which shows phonology as a mostly overlapping between SSD and DLD and other areas, such as articulation, outside this. Later in the review it appears you will only look at phonological types of SSD. I think this is probably a good idea, but it needs to be rationalised in the introduction. I would also like to know if you will consider inconsistent phonological disorder. Similarly, the introduction needs to clarify if you are mostly concerned with DLD and SSD of unknown origin and not associated with known biomedical conditions, and again rationalise this choice in the introduction. It would be useful to define oral vocabulary (receptive, expressive, both?) and comprehensibility- which may differ from intelligibility- in the introduction. In the method I think it would be useful to be clearer about what you will include as a measure of “comprehensibility”. I agree this is likely important to parents, but most SSD intervention studies do not measure this directly. Many also do not measure intelligibility- they often use measure such as percentage consonants correct which, although potentially related to intelligibility- again is a not a direct measure. Presumably what will be most useful is to include any study which measures changes in speech production.
---

	Good luck with the review, I think it will be a really useful piece of work Best Wishes, Joanne Cleland
--	--

REVIEWER	Lisa Goffman The University of Texas at Dallas, Speech, Language, Hearing
REVIEW RETURNED	26-Feb-2023

GENERAL COMMENTS	The co-occurrence of DLD and SSD is frequent and the overarching aim of this work to determine an effective and theoretically motivated approach to intervention of these children, is of high significance. There is much co-occurrence of these two disorders, and often clinicians and researchers focus on the aspect that they are most familiar with, not measuring the other. Thus, the proposed work is timely and important. The focus on oral vocabulary is not clearly justified. Smaller vocabulary size is a marker of late talkers, but not of DLD. The age range (2 to 5 years) is wide and DLD may not yet be identifiable in the earlier years. My largest concern relates to the sole reliance on oral vocabulary as a marker of DLD. This needs to be justified or altered, in my view. The methodological approach to the systematic review generally appeared sound. Theoretically, indeed there are accounts that focus on phonology and semantics related to the lexicon. Therefore, the “lexical restructuring” account is highly relevant, as is the multi-level speech processing model. The inclusion of these accounts provides a strong theoretical basis for the proposed work. The section on current interventions associated with “Child Talk” is extremely vague. Recasts and minimal pairs are mentioned; otherwise, no specifics are included. It would strengthen this section to include some specific candidate approaches. What specific approaches may be hypothesized to align with the theories proposed as underpinning, including the lexical restructuring hypothesis, psycholinguistic models, and neural models? How will approaches found in the literature be aligned with these theories? As mentioned above, the work focused on oral vocabulary and intelligibility. I am not convinced that these are optimal indices of DLD and SSD. Especially of DLD, which morphosyntax and perhaps phonology and word form learning are especially implicated. The ages included are 2;0 to 5;11. The measures used to classify the children as DLD and SSD are not sensitive and specific (e.g., the Preschool Language Scale) and are not evaluated for their quality. The younger children in this age range may be late talkers, and not have DLD. Overall, the exclusive emphasis on vocabulary requires justification as associated with DLD. Morphosyntactic measures are explicitly excluded. Regarding SSD, intelligibility shows substantial developmental change between 2;0 and 5;11. Two year olds are not expected to be intelligible. Thus, more specificity and justification would benefit this work. It may be useful to incorporate meta-bias information on the Data Extraction Form (many of the studies likely will not include some of the important information). Also perhaps the quality of the measures used to identify and categorize the participants should be included. Even in the face of these critiques, this is interesting and important
--

	work and the general methodological approach is sound.
--	--

REVIEWER	Karla K. McGregor Boys Town National Research Hospital
REVIEW RETURNED	01-Mar-2023

GENERAL COMMENTS	This is a thoughtfully designed protocol that will result in a systematic review that I and, more importantly, many clinicians will want to read. The adherence to state-of-the-art guidelines and the PPIE approach to generating the questions are also commendable. My only quibbles concern some lack of methodological details, as described below. As long as the outcomes of children with language or speech problems are provided, will studies where classroom teachers are the interventionists be included (I do think some 5-year-olds will be in kindergarten classrooms, yes)? Teachers do a lot of vocabulary instruction. Will 'intervention' include lab-based training? There are many studies designed to understand the nature of word-learning problems among children with DLD by observing their response to the training of new words, for example. These might have implications for clinical interventions, but they are not directly designed to evaluate an intervention approach. I can see reasons not to include these, but it would be useful to state clearly whether they will or will not be included. Will you distinguish studies designed to evaluate 'effectiveness' from those designed to evaluate 'efficacy'? You will extract dosage information (per Warren et al.) from each study, as is appropriate. But it is not clear to me whether dosage in the various ways that Warren measures it will capture how the speech and word treatments are packaged in studies where both are included. I refer you to this work: Tyler, A. A., Lewis, K. E., Haskill, A., & Tolbert, L. C. (2003). Outcomes of different speech and language goal attack strategies. Journal of speech, language, and hearing research: JSLHR, 46(5), 1077-1094. https://doi.org/10.1044/1092-4388(2003/085) I encourage you to extract and summarize the exact outcomes (e.g. accuracy of word imitation, spontaneous production, comprehensibility to whom [SLP, parent, unfamiliar judge]). I think that is your intention but I not certain. Although not specifically stated as an objective, this review will likely reveal gaps in the knowledge base and in the quality of the knowledge base we do have. I encourage you to include this information.
--

VERSION 1 – AUTHOR RESPONSE

Reviewer: 1

2. Dr. Joanne Cleland, University of Strathclyde

Comment	Other comments
9. In the intro I think it would be helpful to be clearer if the review will consider all types of SSD. You could	Pages 3-4, page 7, page 11 and extraction

refer to the Catalise Venn diagram which shows phonology as a mostly overlapping between SSD and DLD and other areas, such as articulation, outside this. Later in the review it appears you will only look at phonological types of SSD. I think this is probably a good idea, but it needs to be rationalised in the introduction. I would also like to know if you will consider inconsistent phonological disorder.	form Thank you for this suggestion, We will be including all types of phonological SSDs. Further detail has been added, with reference to the Catalise Venn diagram. Type of phonological SSD will also be included in the data extraction form, for further commenting in the results paper if relevant to the findings.
10. Similarly, the introduction needs to clarify if you are mostly concerned with DLD and SSD of unknown origin and not associated with known biomedical conditions, and again rationalise this choice in the introduction.	Pages 3-4 Thank you- this did need to be clearer. We have added some sentences to explain that we will be looking at phonological SSDs which are idiopathic, due to their known overlap with DLD (which is also idiopathic).
11. It would be useful to define oral vocabulary (receptive, expressive, both?) and comprehensibility- which may differ from intelligibility- in the introduction.	Page 3 We have also made some additions to the introduction regarding the difference between intelligibility/comprehensibility and that by looking at oral vocabulary we mean expressive.
12. In the method I think it would be useful to be clearer about what you will include as a measure of “comprehensibility”. I agree this is likely important to parents, but most SSD intervention studies do not measure this directly. Many also do not measure intelligibility- they often use measure such as percentage consonants correct which, although potentially related to intelligibility- again is a not a direct measure. Presumably what will be most useful is to include any study which measures changes in speech production.	Page 8 Yes, thank you for pointing this out. There has been much deliberation on this. The original intention was to look at PCC as a proxy for comprehensibility. However, after reviewing the literature for PCC (and alternate measures of speech accuracy), it was felt that speech comprehensibility is about more than speech accuracy, and cannot be fully captured by PCC. Therefore, the closest proxy deemed suitable was intelligibility.

	There are SSD studies which do not explicitly target intelligibility/comprehensibility but may still have it as an additional measure, and our inclusion criteria has been designed to still capture these studies. We suspect one of the main measures of comprehensibility will actually be the intelligibility in context scale, although we appreciate that the boundary between intelligibility and comprehensibility within this measure is not clear cut. We do understand however that intelligibility/comprehensibility are not always routinely measured even as a secondary outcome. We will comment on this in the results paper should the findings indicate appropriateness for this.
--	---

Reviewer: 2

Prof. Lisa Goffman , The University of Texas at Dallas

Comment	Response
13.The focus on oral vocabulary is not clearly justified. Smaller vocabulary size is a marker of late talkers, but not of DLD. The age range (2 to 5 years) is wide and DLD may not yet be identifiable in the earlier years. My largest concern relates to the sole reliance on oral vocabulary as a marker of DLD. This needs to be justified or altered, in my view.	Page 4 and page 7 Thank you for flagging this up. The wording of this has been adjusted to hopefully make our rationale clearer. Specifically, that due to challenges within diagnosing DLD in very young children, and not wanting to exclude intervention techniques of potential relvence, we will be including children who have features of DLD (including late talkers) who do not necessarily have a diagnosis yet.
14.The section on current interventions associated with “Child Talk” is extremely vague. Recasts and minimal pairs are mentioned; otherwise, no specifics are included. It would strengthen this section to include some specific candidate approaches. What specific approaches may be hypothesized to align with the theories proposed as underpinning, including the lexical restructuring hypothesis, psycholinguistic models, and	Page 5 and page 12 This has provided great food for thought, thank you. Although the final results paper will go into substantially more detail on this, we understand that making some additions here within the protocol would be

neural models? How will approaches found in the literature be aligned with these theories?	beneficial. Additions have been made to explicitly connect some of the findings from child talk to the presented theories of potential relevance, as well as some further elaboration in the data synthesis section. In particular, we have discussed how lexical restructuring may occur as a result of language modelling, and how 'building things into play' will capitalise on the child's attention/motivation, thus providing an optimal environment for speech and language learning in very young children.
15.As mentioned above, the work focused on oral vocabulary and intelligibility. I am not convinced that these are optimal indices of DLD and SSD. Especially of DLD, which morphosyntax and perhaps phonology and word form learning are especially implicated.	Page 3, page 5, page 12 This comment is very much appreciated, we are aware that there is more literature around morphosyntactic/word form aspects of DLD and the overlap with SSD. Interestingly, the initial DLD outcome for this body of intervention development work was actually related to morphosyntax, not oral vocabulary. However, when the pre-study patient and public involvement/ engagement work was conducted, comprehensibility and oral vocabulary were the areas that parents and clinicians said would have the most positive impact for a young child accessing a dual SSD/DLD intervention. Clinicians fed back that within the early years, particularly for children with more severe features of DLD, they rarely get to work on morphosyntax as the child does not have enough vocabulary yet. They also reported that building up oral vocabulary would have more of a functional impact for the child. As this programme of work progresses and the intervention currently being developed

	is being tested, it will be interesting to see if there are any effects on morphosyntax once oral vocabulary/speech comprehensibility have improved. There could also be scope for a separate intervention to be developed/adapted, perhaps for slightly older children or children with at a more advanced level of their language learning, which specifically targets outcomes in morphosyntax (perhaps building on the valuable work of Prof. Ann Tyler).
16.The ages included are 2;0 to 5;11. The measures used to classify the children as DLD and SSD are not sensitive and specific (e.g., the Preschool Language Scale) and are not evaluated for their quality. The younger children in this age range may be late talkers, and not have DLD.	Extraction form Yes we completely agree and are mindful that diagnosis of DLD in the early years is currently a challenge for both clinicians and researchers. Please see our response to 13 re: looking at children with features of DLD (which includes late talkers). We have also taken on board your later comment about incorporating further assessment details into the extraction form.
17.Overall, the exclusive emphasis on vocabulary requires justification as associated with DLD. Morphosyntactic measures are explicitly excluded. Regarding SSD, intelligibility shows substantial developmental change between 2;0 and 5;11. Two year olds are not expected to be intelligible. Thus, more specificity and justification would benefit this work.	Re: emphasis on vocabulary- please see the our responses to points 13 and 15. Page 3 Comprehensibility/intelligibility in very young children – yes, we definitely see what you mean- there is a lot of change during this time. Although some level of intelligibility/comprehensibility is ‘typical’. We have added some additional information into the introduction regarding this, based on an interesting recent study https://doi.org/10.1044/2021_JSLHR-21-00142

	Page 7 Although this review forms part of a larger intervention development study for children aged 3-4 years, we have kept the age range for our review wider than this as we do not want to miss intervention techniques of potential relevance. We do acknowledge however that we are more likely to find studies including intelligibility/comprehensibility as an outcome at the older end of our specified age range.
18. It may be useful to incorporate meta-bias information on the Data Extraction Form (many of the studies likely will not include some of the important information). Also perhaps the quality of the measures used to identify and categorize the participants should be included.	Page N/A- extraction form Thank you for this helpful suggestion. This has been added to the extraction form. Although quality of measures used to identify the children is not a focus of the study, it is appreciated that having this information could help provide further context to the overall relevance of the included papers.

Reviewer: 3

Dr. Karla K. McGregor, Boys Town National Research Hospital

Comment	Response
19. As long as the outcomes of children with language or speech problems are provided, will studies where classroom teachers are the interventionists be included (I do think some 5-year-olds will be in kindergarten classrooms, yes)? Teachers do a lot of vocabulary instruction.	Page 8 Thank you for pointing this out-yes it would overlap with the kindergarten age. We will be including studies where teachers are the interventionists, and have added this to our wording to make this clearer.
20. Will 'intervention' include lab-based training? There are many studies designed to understand the nature of word-learning problems among children with DLD by observing their response to the training of new words, for example. These might have implications for clinical interventions, but they are not directly designed to	Page 7 and page 12 Great point- we did not make this explicit. We are indeed including such lab-based training, where the 'intervention' is the child

evaluate an intervention approach. I can see reasons not to include these, but it would be useful to state clearly whether they will or will not be included.	being trained to learn new words (as long as the other inclusion criteria are met e.g., there is an appropriate comparator, and an 'intervention' has taken place). We acknowledge that these words are not learnt in 'real life', and therefore information regarding the location, wider context etc will be limited. Still, we would not want to rule out valuable information regarding intervention techniques which these studies may provide. Data from such studies will be used differently within the data syntheses- we have adjusted the wording within this section also to reflect this.
21. Will you distinguish studies designed to evaluate 'effectiveness' from those designed to evaluate 'efficacy'?	Page 7 and page 12 Building from our comment above- we took studies on efficacy to refer to lab-based training studies, which are highly controlled. We will not be excluding studies which are purely efficacy based, as long as the inclusion criteria are met. However, we have reflected on your comment and made this clearer in the protocol, altering the wording for our data synthesis section.
22. You will extract dosage information (per Warren et al.) from each study, as is appropriate. But it is not clear to me whether dosage in the various ways that Warren measures it will capture how the speech and word treatments are packaged in studies where both are included. I refer you to this work: Tyler, A. A., Lewis, K. E., Haskill, A., & Tolbert, L. C. (2003). Outcomes of different speech and language goal attack strategies. Journal of speech, language, and hearing research: JSLHR, 46(5), 1077-1094. https://doi.org/10.1044/1092-4388(2003/085)1094.	Page 11 and extraction form Great point. Although we may be unlikely to find many studies which measure both oral vocabulary and speech comprehensibility together, we do not know for certain. Although we are using Warren et al. as a framework for dosage extraction, our aim is to capture all dosage information given in the included papers. For both oral vocabulary/speech comprehensibility intervention studies, this would include any additional information on dosage within a dual intervention which either integrates speech/language content or alternates it.

	I also think this links in with your final comment (24) about potential gaps in the knowledge base, as (unlike in Prof. Tyler's study) other studies may not contain this level of detail. Like you said, it is important that we comment on these gaps when observed.
23.I encourage you to extract and summarize the exact outcomes (e.g. accuracy of word imitation, spontaneous production, comprehensibility to whom [SLP, parent, unfamiliar judge]). I think that is your intention but I not certain.	Page N/A- extraction form Thank you. Yes it is but we have now added this detail in to our extraction form to make it clearer.
24.Although not specifically stated as an objective, this review will likely reveal gaps in the knowledge base and in the quality of the knowledge base we do have. I encourage you to include this information.	Page 12 Yes definitely, thank you for pointing out this out- we will include this information in the results paper and have clarified this in the protocol.

VERSION 2 – REVIEW

REVIEWER	Joanne Cleland University of Strathclyde, Psychological Sciences and Health
REVIEW RETURNED	20-Mar-2023

GENERAL COMMENTS	The authors have done a good job addressing the reviewer's comments. I particularly wish to thank the reviewers for the way they set out their changes in their appended word document. It is my view that the paper is substantially improved and that this review will make a very useful addition to the literature. I look forward to reading the final review. One small point is that the authors do in several places highlight that DLD and phonological SSD are both of unknown origin/idiopathic and therefore distinct from the other SSDs of known causes. Two minor points of clarification on this: articulation disorder and CAS (sometimes) are also idiopathic- I suspect the authors know this, but highlighting dysarthria as being of known cause does somewhat falsely lead the reader to believe that all the non-phonological subtypes of SSD are of known cause. Secondly, it is possible that in the future causes (likely more than one) will be identified for both DLD and SSD. I will leave it to the authors to decide whether they clarify this in the paper.
--

REVIEWER	Lisa Goffman The University of Texas at Dallas, Speech, Language, Hearing
REVIEW RETURNED	20-Mar-2023

GENERAL COMMENTS

Consistent with my prior review, I think that the proposed work is generally of high significance. As well expressed by the authors, children with co-occurring deficits in language (DLD) and phonology (SSD) are at great risk for negative outcomes. Further, there are few efficacious treatment approaches addressing both language and speech. Thus, the overarching aim of this work is important.

Certainly, especially in early development, there is much evidence that the acquisition of sounds and words is highly interactive. Further, there is a gap in the literature, with few theoretically informed interventions. This work is designed as a first step in designing a new and effective intervention for children with co-occurring DLD and SSD. This is a worthy goal.

A few concerns remain.

I appreciated the inclusion of play, but it was unclear how it was specifically incorporated into the analytic approach (e.g., the data extraction form).

Also, it may be helpful to include more specifics about the scope of what is meant by vocabulary and comprehensibility.

The focus on co-occurring features of phonology and words is a strength.

The authors have clarified why it is reasonable to include late talkers. However, as in my prior review, I remain concerned that DLD outcome is defined exclusively in relation to oral vocabulary—perhaps it would be useful to define more specifically what oral vocabulary is and how it is measured. As stated below, finer definitions related to comprehensibility would also be helpful. I remain unconvinced that standard measures of vocabulary relate to DLD.

Comprehensibility is an important construct. However, to my knowledge, it is not the standard approach to measuring change in children with SSD. I am concerned about finding a sufficient number of studies that use this measure (or that of intelligibility, which is quite slippery to measure) and wonder if the net should be cast more widely at this point in the development of this work (e.g., PCC, PPC, variability, performance on an articulation/phonology test)? The SSD constructs may need to be broader. This seems supported by the search strategy data showing that only two papers (out of 41) addressed outcome of comprehensibility or intelligibility. Camarata should be cited in discussing comprehensibility. Others have looked at the speech and language interface and should probably be cited.

I appreciated that the authors addressed why vocabulary was selected as the primary language measure, based on parent and clinician input. However, it is becoming increasingly clear that measures of static vocabulary do not adequately index language learning. For example, the value of the "million word gap" as a relevant index of language development is being questioned and may be considered culturally biased. A core tenet of language acquisition is that little input is obligated to learn language, and that children learn in the face of multiple kinds of input (except of course in the case of DLD; but here many children diagnosed with DLD perform within expected levels on measures of vocabulary). Vocabulary tests are especially vulnerable to these input factors--e.g., the specific types of words children are exposed to. Even in the

	face of clinician and parent input, I am not confident that vocabulary is empirically supported as an optimal intervention approach or outcome measure. As with my first review, I remain unconvinced that there is an evidence base for a focus on vocabulary as an index of DLD. That said, I find value in this study. A couple of small points: This is a minor point, but phonology extends beyond individual sounds, and incorporates sound patterns and prosody (e.g., assimilation/harmony and weak syllable deletion patterns). I'm sympathetic with the inclusion of meaningful interactions in language learning, but why is this related to the neural basis for speech and language development?
--	---

REVIEWER	Karla K. McGregor Boys Town National Research Hospital
REVIEW RETURNED	21-Mar-2023

GENERAL COMMENTS	I reviewed the original submission and I find this version to satisfy the minimal concerns that I had.
--

VERSION 2 – AUTHOR RESPONSE

Reviewer: 1

Dr. Joanne Cleland, University of Strathclyde

Thank you once again for taking the time to read and review our revised protocol. We are pleased that you largely found our amended paper satisfactory and look forward to sharing the findings with you.

Comment	Other comments
1. One small point is that the authors do in several places highlight that DLD and phonological SSD are both of unknown origin/idiopathic and therefore distinct from the other SSDs of known causes. Two minor points of clarification on this: articulation disorder and CAS (sometimes) are also idiopathic- I suspect the authors know this, but highlighting dysarthria as being of known cause does somewhat falsely lead the reader to believe that all the non-phonological subtypes of SSD are of known cause. Secondly, it is possible that in the future causes (likely more than one) will be identified for both DLD and SSD. I will leave it to the authors to decide whether they clarify this in the paper.	Page 4 Thank you for highlighting this, we have amended the sentence saying that non phonological SSDs usually have no known cause, to specifically state that articulation disorder/CAS can be idiopathic.

Reviewer: 2

Prof. Lisa Goffman , The University of Texas at Dallas

Thank you for your further comments, all of which we appreciate. Our responses are outlined below.

Comment	Response
2. I appreciated the inclusion of play, but it was unclear how it was specifically incorporated into the analytic approach (e.g., the data extraction form).	Page: 6 When extracting intervention techniques, we will be looking at what type of activities these will be in; play will be one of these activities (as well as other activities such as shared book reading). We have reworded this slightly to clarify.
3. Also, it may be helpful to include more specifics about the scope of what is meant by vocabulary and comprehensibility. However, as in my prior review, I remain concerned that DLD outcome is defined exclusively in relation to oral vocabulary—perhaps it would be useful to define more specifically what oral vocabulary is and how it is measured. As stated below, finer definitions related to comprehensibility would also be helpful. I remain unconvinced that standard measures of vocabulary relate to DLD.	Pages 2,8 Comprehensibility- we have taken the definition as established by Pommee et al. (page 2). We have added a couple of sentences to expand on this further within the outcomes section (page 8), based on your comments in point 7. These comments acknowledge the valuable work of Prof. Camarata, as well as other leaders in the field such as Prof. Sharynne McLeod (developer of the Intelligibility in Context Scale). For the purpose of this review we will include a range of measures for oral vocabulary and have added some additional examples on page 8- e.g. parental report instruments and token type ratios. We have further elaborated on this issue in addressing your next point.

4. I appreciated that the authors addressed why vocabulary was selected as the primary language measure, based on parent and clinician input. However, it is becoming increasingly clear that measures of static vocabulary do not adequately index language learning. For example, the value of the "million word gap" as a relevant index of language development is being questioned and may be considered culturally biased. A core tenet of language acquisition is that little input is obligated to learn language, and that children learn in the face of multiple kinds of input (except of course in the case of DLD; but here many children diagnosed with DLD perform within expected levels on measures of vocabulary). Vocabulary tests are especially vulnerable to these input factors--e.g.. the specific types of words children are exposed to. Even in the face of clinician and parent input, I am not confident that vocabulary is	We understand your concerns relating to vocabulary measures and what they really tell us, particularly, specific types of words children are exposed to/risk of cultural bias. However, we feel that including a range of outcome measures for vocabulary use within this review enables the exploration of the theory that underpins intervention techniques for oral vocabulary, and their relation to intervention techniques for phonological SSD. Your point has definitely highlighted how carefully we need to treat differing measures of vocabulary, with further commentary in the discussion section. As mentioned on page 13, we are lucky to have an clinical EDI expert and early years' bilingual educational family support worker as part of our project steering group, who can advise on the interpretation of the data from the varied outcome measures. Additionally we have found your comment re: focus on vocabulary as a feature of DLD, really interesting. For the purpose of the intervention being developed, in part as a result of this review, we have to balance evidence with the needs of the families who we are aiming to support. We appreciate the importance of other features of DLD, but for the purpose of an intervention for this age group, oral vocabulary has been prioritised by clinicians/families. We know that such prioritisation by key stakeholders is advised in order to facilitate successful implementation of interventions being developed further down the line https://bmiopen.bmj.com/content/9/8/e029954 We have referenced CATALISE; where limited vocabulary is highlighted as a key feature at age 2-3 years, and issues related to word linking are highlighted for ages 3 to 4 and 4 to 5 years: https://journals.plos.org/plosone/article?id=10.1371/journal.pone.0158753#sec006 An increase in vocabulary has the potential to provide a foundation for such word linking to take place: https://www.tandfonline.com/doi/full/10.1080/02699200410001716165 This corroborates what clinicians told us within our patient and public involvement/engagement work; they would not work on extending word linking/utterance length until a child has sufficient vocabulary to do this.
---	---

empirically supported as an optimal intervention approach or outcome measure. As with my first review, I remain unconvinced that there is an evidence base for a focus on vocabulary as an index of DLD. That said, I find value in this study.	
This is a minor point, but phonology extends beyond individual sounds, and incorporates sound patterns and prosody (e.g., assimilation/harmony and weak syllable deletion patterns).	Thank you for the reminder. We will certainly be mindful of this in future communications.
6. I'm sympathetic with the inclusion of meaningful interactions in language learning, but why is this related to the neural basis for speech and language development?	Page 5 We have added a sentence to make this more explicit. In the Romeo et al (2018) study, they found that Broca's area of the brain became activated when children were engaged in meaningful back and forth interactions-this had more of an impact than the number of words heard. It's a fascinating paper: https://journals.sagepub.com/doi/pdf/10.1177/0956797617742725
7. Comprehensibility is an important construct. However, to my knowledge, it is not the standard approach to measuring change in children with SSD. I am concerned about finding a sufficient number of studies that use this	Thank you for raising this point, which was also raised by reviewer 1 in their first round of feedback. There has been much deliberation on this. The original intention was to look at PCC as a proxy for comprehensibility. However, after reviewing the literature for PCC (and alternate measures of speech accuracy), it was felt that speech comprehensibility is about more than speech accuracy, and cannot be adequately captured by PCC. Therefore, the closest proxy deemed suitable was intelligibility. There are SSD studies which do not explicitly target intelligibility/comprehensibility but may still have it as an additional measure, and our inclusion criteria has been designed to still capture these studies. We

measure (or that of intelligibility, which is quite slippery to measure) and wonder if the net should be cast more widely at this point in the development of this work (e.g., PCC, PPC, variability, performance on an articulation/phonology test)? The SSD constructs may need to be broader. This seems supported by the search strategy data showing that only two papers (out of 41) addressed outcome of comprehensibility or intelligibility. Camarata should be cited in discussing comprehensibility. Others have looked at the speech and language interface and should probably be cited.	suspect one of the main measures of comprehensibility will actually be the intelligibility in context scale, although we appreciate that the boundary between intelligibility and comprehensibility within this measure is not clear cut. We do understand however that intelligibility/comprehensibility are not always routinely measured even as a secondary outcome. We will comment on this in the results paper should the findings indicate appropriateness for this. Page 8 Thank you for reminding us about Prof. Stephen Camarata's work in the area. Interestingly, it was actually one of his papers (with Prof. Paul Yoder) which we identified for potential inclusion from the original systematic review. We have added extra detail regarding measuring speech comprehensibility and intelligibility, based on his work, as well as the work of Prof. Sharynne McLeod. We clarify that we will include parent ratings scales alongside objective measures, as both still relate to functional communication.
--	--

Reviewer: 3

Dr. Karla K. McGregor, Boys Town National Research Hospital

N/A.

Thank you once again for taking the time to read and review our revised protocol. We are pleased that you found the amendments satisfactory and look forward to sharing the findings with you.

VERSION 3 – REVIEW

REVIEWER	Lisa Goffman The University of Texas at Dallas, Speech, Language, Hearing
REVIEW RETURNED	23-May-2023
GENERAL COMMENTS	As in my prior reviews, I find the proposed work of high significance. I continue to have some concerns about vocabulary as the key measure, though I realize, as well articulated by the authors, that this is an area of concern for families. Words also serve as a focus of speech sound intervention. And most certainly word form learning

	is a key area of deficit in DLD (e.g., McGregor's findings in children and adults). I do think it would be important, in the final product, to include more work about speech and language interactions in typical and atypical development. And I remain uncertain that semantic measures of vocabulary will be substantive markers of DLD. All of that said, the authors have addressed key issues and I support their proceeding with this important work. I look forward to the results.
--	--